# OpenReview forum: "DetailMaster: Can Your Text-to-Image Model Handle Long Prompts?"
_ICLR.cc/2026/Conference — ICLR 2026 Conference Withdrawn Submission_

### Official Review · Reviewer_YsLh · 2025-10-19

**Soundness:** 2
**Presentation:** 3
**Contribution:** 3
**Rating:** 4
**Confidence:** 3

**Summary:**

The paper introduces DETAILMASTER, the first benchmark for evaluating T2I models on long detail-rich prompts, focusing on four dimensions: Character Attributes, Structured Character Locations, Multi-Dimensional Scene Attributes, and Spatial/Interactive Relationships.

It identifies T2I models’ flaws (compositional reasoning issues, attribute leakage) in long prompts, sets a standard for evaluation, and enables progress in professional applications like industrial prototyping hindered by poor detail fidelity.

**Strengths:**

1. It fills the gap with long-prompt T2I evaluation by proposing DETAILMASTER— they claim as the first benchmark (actually concurrent with [1] TIT-Score) with avg 284.89-token prompts (vs. existing <100-token ones) and 4 targeted dimensions (e.g., Structured Character Locations).

2. It reveals critical T2I flaws (prompt-length accuracy degradation, backbone limits) and guides research (prioritize long-prompt training over context windows).

[1] TIT-Score: Evaluating Long-Prompt Based Text-to-Image Alignment via Text-to-Image-to-Text Consistency (Arxiv Oct 3rd.)

**Weaknesses:**

**1. Incomplete Failure Mechanism Validation:** It attributes poor performance to "encoder grammar flattening" and "diffusion attribute leakage" but lacks causal tests. For instance, no ablation comparing T5 (grammar-aware) vs. CLIP (flat) encoders on prompt structure preservation, or attribute leakage tracking via attention visualization. Adding such experiments would confirm root causes and guide model improvements. Instead of proposing a benchmark, which is more like an effort job, readers may want to know more insights from the huge benchmark experiments.

For example:

(1) what is the reason to make the T2I models behave diversely? The training data, training scheme or text encoder differences?

(2) Any potential to solve the problem with your proposed benchmark?

(3) How are the unified models (BAGEL, Blip3-o, Janus, Janus-pro, Janus-flow, etc.) perform on your benchmark?

(4) Would better LLM/VLM as the encoder benefit the generation?

**2. High Evaluation Computational Barrier:** The MLLM-based evaluation requires 20GB–39GB GPU VRAM and 10+ hours per run, excluding small teams. It could adopt lightweight MLLMs (e.g., Qwen2.5-VL-2B) fine-tuned on its annotation data, or distill the evaluator into a smaller model. This would reduce resource needs while preserving accuracy.

**Questions:**

refer to the weaknesses.

---

> ### Author Response · Authors · 2025-11-20
> **Responses to Reviewer YsLh [Part 1/7]**
>
> Dear Reviewer YsLh, we are immensely grateful for your constructive review! We make the following detailed responses point by point to address your comments with substantial experimental support, hoping to clarify our paper and encourage you to lean toward its acceptance. The additional experimental results and discussions have also been added to the revised version of our paper, as indicated by the blue text.
>
> ---
>
> # W1.1 A More Detailed Analysis of Flattening Complex Grammatical Structures and Attribute Leakage
>
> Thank you for pointing out the need for greater clarity regarding the analyses supporting  "flattening of grammatical structures" and "attribute leakage". We acknowledge that Section 4 does not contain explicitly titled subsections dedicated to these points, and we have addressed this by **newly adding a dedicated subsection in the revised version** to explicitly connect our experimental findings to these broader conclusions.
>
> In our original paper, the evidence supporting these two claims has been embedded within the following experimental results:
>
> - **Regarding encoders flattening complex grammatical structures (Evidence from Section 4.2.3 & Appendix M):** Our analyses in Section 4.2.3 and Appendix M show that "Character Locations" and "Entity Relationships" are among the most challenging tasks. These tasks require a nuanced understanding of *complex spatial instructions, inter-object relationships, and sentential hierarchies.* The models' failures in these areas strongly suggest that **their encoders struggle to parse such structured, non-linear descriptions**, instead **treating them as "flattened" sets of features**. Furthermore, our error analysis in Appendix M reveals dominant failure modes, including: 1) conflicts between spatial instructions and the model's real-world positional priors, and 2) difficulties in rendering complex character interactions. These failure cases directly reflect **the model's fundamental difficulty in comprehending instructions involving complex grammatical relations** (e.g., prepositional phrases, subordinate clauses).
> - **Regarding "attribute leakage" in diffusion models under detail-intensive conditions (Evidence from Section 4.3 & Appendix N):** One of our key findings is *the consistent decline in the accuracy of all evaluated models as prompt token length increases*. This is not solely due to character omission. As further detailed in Appendix N, **even for successfully generated characters, the fidelity of their attributes decreases as the prompt lengthens**. This is direct evidence of "attribute leakage": when the model is overwhelmed by a detail-intensive prompt, it fails to strictly "bind" specific attributes (e.g., colors, clothing, actions) to the correct objects, resulting in attributes being omitted, distorted, or incorrectly applied to other objects.
>
> Thank you for prompting us to clarify this. In the **revised paper**, we **have added a paragraph at the end of Section 4** to more explicitly link these specific experimental findings to our conclusions (**Section 4.4**).
>
> (To be continued.)

---

> ### Author Response · Authors · 2025-11-20
> **Responses to Reviewer YsLh [Part 2/7]**
>
> (Continued from the previous page. W1.1)
>
> # W1.2 "What is the reason to make the T2I models behave diversely?"
>
> Thank you for the insightful question, which strikes at the core of our investigation. The diverse behaviors of T2I models when faced with long, detail-intensive prompts are driven by a complex interplay of the factors mentioned: the text encoder, the model architecture, the training data, and the training scheme. Our work with DetailMaster is specifically designed to expose and quantify the effects of these elements.
>
> Based on our evaluations (presented in Table 2 and analyzed in Section 4.2), we can elaborate on the contribution of each factor:
>
> - **Text Encoder Differences.** The choice of text encoder fundamentally determines a model's capacity to process long prompts. Our results clearly distinguish between two primary classes of encoders: (1) **CLIP-based Encoders** (e.g., in SD1.5, SD-XL). These models are severely hampered by CLIP's inherent 77-token limit. *Any information beyond this hard cutoff is simply ignored*, leading to catastrophic failures in prompt adherence, as evidenced by their consistently low scores across all four evaluation dimensions in Table 2. (2) **LLM-based Encoders** (e.g., T5 in DeepFloyd IF, SD3.5, FLUX). Models that leverage powerful LLMs like T5 as their text encoder *possess a significantly larger context window*. This architectural choice is **a prerequisite for handling long prompts** and directly correlates with the substantial performance leap observed between SD-XL and models like SD3.5. Additionally, during the rebuttal phase, we evaluate the performance of *Qwen-Image*, which utilizes Qwen2.5-VL as its text encoder and demonstrates enhanced capabilities in understanding long prompts.
> - **Training Data and Scheme.** This is arguably the most critical factor, and our findings provide strong evidence for it. Simply having a capable encoder is not enough, the model must be explicitly trained to utilize it effectively on complex inputs. Our results find that: (1) **Short Prompt Bias.** As stated in our introduction, most classic T2I models are predominantly trained on datasets with short, simple captions (e.g., COCO, CC12M). *This ingrains a bias towards generating images from concise descriptions but leaves them unprepared for parsing the complex compositional and relational grammar of long prompts.* (2) **Long-Prompt Training.** Our analysis of long-prompt optimized models in Section 4.2.2 provides a direct answer. We compare DeepFloyd IF (which uses an LLM-based encoder but is trained on conventional data) with ParaDiffusion (which also uses an LLM-based encoder but is trained with long prompts). ParaDiffusion consistently outperforms DeepFloyd IF across most metrics. This demonstrates that **explicit training on detail-rich, long prompts is more impactful than merely increasing the token capacity of the encoder**. Models like ELLA and LongAlign, which also incorporate long-prompt training schemes, similarly show marked improvements over their baselines despite architectural constraints.
> - **Architectural and Methodological Innovations.** Beyond the encoder and data, novel architectural refinements play a crucial role in mitigating specific failure modes like attribute leakage and spatial reasoning errors: (1) **Iterative Decomposition.** Models like LongAlign and LLM Blueprint employ a "divide and conquer" strategy. *They decompose the long prompt into manageable semantic chunks or extract key object layouts before generation.* This structured approach helps preserve compositional integrity and leads to notable gains, particularly in challenging tasks like "Character Locations." (2) **Enhanced Architectures.** The superior performance of SOTA models like SD3.5 and FLUX over older T5-based models like DeepFloyd IF is not just due to better data, but also to superior model architecture. *The advancements in the diffusion model itself likely improve its ability* to bind attributes to objects and prevent the "attribute leakage" we hypothesize in our conclusion.
>
> (To be continued.)

---

> ### Author Response · Authors · 2025-11-20
> **Responses to Reviewer YsLh [Part 3/7]**
>
> (Continued from the previous page. W1.2 part1)
>
> ## **Ablation Study with a Controlled Architecture for Variable Isolation**
>
> To better compare the performance gains brought by various long-prompt optimization strategies, during the rebuttal phase, **we conduct a more controlled ablation study using a single model architecture to isolate variables**.
>
>
>
> ### **Setup for New Experiments**
>
> We modify SD-XL by integrating a T5 text encoder alongside the original CLIP encoders. To fuse the features from the T5 encoder, we adopt a Timestep-Aware Semantic Connector, similar to the mechanism used in ELLA. This structure allows the model to leverage the extended prompt limit and rich semantic representation of T5 *without discarding the capabilities learned from CLIP features.* For efficiency, we keep the weights of the original SD-XL UNet, CLIP, and T5 frozen, *only training the new semantic connector.* As for the training data, we curate a dataset of 2M image-detailed caption pairs from the DOCCI dataset and the Localized Narratives dataset.
>
> We design four distinct experimental settings to systematically evaluate the impact of prompt limit and training prompt length:
>
> 1. **77 Token Limit + Short-Prompt Training**.
> 2. **512 Token Limit + Short-Prompt Training**.
> 3. **77 Token Limit + Long-Prompt Training**.
> 4. **512 Token Limit + Long-Prompt Training**.
>
> Herein, **the short prompts** are generated by using Qwen2.5-VL-7B-Instruct to compress the original detailed captions to a length of 30 tokens or less, preserving the core semantic content.
>
>
>
> ### **Results and Analysis**
>
> We evaluate all four fine-tuned models on our DetailMaster benchmark. The results are presented in the table below:
>
>
>
> | Model                              | Object | Animal | Person | Character Locations | Background | Light | Style | Entity Relationships |
> | ---------------------------------- | ------ | ------ | ------ | ------------------- | ---------- | ----- | ----- | -------------------- |
> | 77 Limit w/ Short-Prompt-Training  | 24.38  | 28.17  | 16.42  | 9.39                | 29.31      | 74.72 | 46.64 | 8.43                 |
> | 512 Limit w/ Short-Prompt-Training | 25.37  | 30.11  | 18.27  | 10.45               | 34.8       | 74.16 | 66.22 | 10.12                |
> | 77 Limit w/ Long-Prompt-Training   | 27.44  | 31.54  | 18.32  | 10.93               | 33.7       | 76.62 | 53.72 | 10.66                |
> | 512 Limit w/ Long-Prompt-Training  | 30.5   | 33.13  | 19.36  | 12.52               | 37.51      | 80.68 | 69.98 | 13.07                |
>
> These controlled experiments yield several key insights that strongly support the original conclusions of our paper:
>
> - **Both Factors Matter:** Comparing the results to the baseline (77 Limit w/ Short-Prompt), both expanding the prompt limit and using long-prompt training lead to performance improvements across most compositional metrics, especially in attribute and location accuracy.
> - **Long-Prompt Training is More Impactful than Token Capacity Alone:** A direct comparison between the second and third setups reveals that the performance gain from training on long, descriptive prompts is more significant than the gain from merely increasing the prompt limit while training on short prompts. For instance, the model trained on long prompts with a 77-token limit outperforms the one trained on short prompts with a 512-token limit in Object, Animal, Person, Character Locations, and Entity Relationships.
> - **Synergistic Effect:** The best performance is achieved when both factors are combined (512 Limit w/ Long-Prompt Training). This result highlights a clear synergistic effect, suggesting that the combination of an expanded prompt limit and dedicated long-prompt training is a critical strategy for improving long-prompt adherence.
>
> **We have newly added this ablation study and its analysis to our revised paper to substantiate our claims more rigorously (Section 4.5)**.
>
> In summary, the diverse performance we observe is not due to a single cause, but a synergistic combination of these three factors. A powerful **text encoder** provides the necessary capacity, explicit **long-prompt training data and schemes** teach the model the skill to leverage that capacity, and advanced **architectures/methodologies** provide the fine-grained control needed for high-fidelity compositional generation. DetailMaster serves as a crucial diagnostic tool for the community, making these distinct contributions and their interplay transparent and measurable.
>
> (To be continued.)

---

> ### Author Response · Authors · 2025-11-20
> **Responses to Reviewer YsLh [Part 4/7]**
>
> (Continued from the previous page. W1.2 part2)
>
> # W1.3 "Any potential to solve the problem with your proposed benchmark?"
>
> Thank you for this forward-looking question. While a benchmark itself is a diagnostic instrument rather than a direct solution, we argue that **DetailMaster is a critical catalyst for solving the long-prompt generation problem by providing the necessary tools for diagnosis, guidance, and training.** Our contribution is designed to directly enable the community to solve this challenge in two key ways:
>
> 1. **Transforming the Problem from Anecdotal to Quantifiable:** Prior to our work, the failure of T2I models on long prompts was a well-known but poorly defined issue. Researchers lacked a standardized way to measure how and why models were failing. DetailMaster transforms this challenge into a concrete, quantifiable research problem.
>
> - - **Granular Diagnosis:** Our four evaluation dimensions (i.e., Character Attributes, Structured Character Locations, Multi-Dimensional Scene Attributes, and Spatial/Interactive Relationships) provide a fine-grained breakdown of model capabilities. For the first time, researchers can move beyond a simple "pass/fail" and precisely identify whether a new architecture is failing at attribute binding, spatial reasoning, or scene composition in long-prompt scenarios.
>   - **Identifying the Hierarchy of Difficulty:** Our analysis reveals a clear "hierarchy of compositional failure" (Appendix M), demonstrating that precise spatial reasoning (Character Locations) is the most challenging task for current models. This insight directs the community's focus toward the most critical bottlenecks, encouraging research into areas like structured spatial representations rather than just general fidelity.
>
> 2. **Enabling Data-Centric Approaches:** By open-sourcing our dataset and data construction process, we empower researchers to **pursue data-centric solutions**. As we propose in our conclusion, scaling our generation pipeline can create massive, high-quality datasets for "**Curriculum-Based and Data-Centric Training**", which is a crucial path toward robust long-prompt T2I models. Additionally, DetailMaster is not just an evaluation set. It is a large-scale, high-quality dataset of 4,116 long prompts (avg. 284.89 tokens) with rich, structured annotations. This dataset can be directly used for fine-tuning existing models or as a component in pre-training new ones to specifically improve their long-prompt adherence and compositional skills.
>
>
>
> In conclusion, while DetailMaster is a benchmark, its potential extends far beyond measurement. It provides the **clarity, guidance, data synthesis pipeline, and standardized framework** necessary for the research community to systematically diagnose, address, and ultimately solve the critical challenge of long-prompt adherence in text-to-image generation.
>
> (To be continued.)

---

> ### Author Response · Authors · 2025-11-20
> **Responses to Reviewer YsLh [Part 5/7]**
>
> (Continued from the previous page. W1.3)
>
> # W1.4 "How are the unified models perform on your benchmark?"
>
> Thank you for the valuable suggestion. To provide a clearer picture of how recent unified models perform on our benchmark, we conduct an evaluation on several advanced unified models, including **Janus, JanusFlow, Janus-Pro, Lumina-Image-2.0, and BAGEL**.
>
> The results are presented in the table below:
>
> | Model   | Object | Animal | Person | Character Locations | Background | Light | Style | Entity Relationships |
> | - | --| - | --| -- | -- | --- | -- | - |
> | Janus-1.3B   | 35.86  | 32.15  | 22.31  | 21.4 | 87.59   | 95.67 | 83.53 | 25.48  |
> | JanusFlow-1.3B  | 31.93  | 37.54  | 16.5   | 17.3 | 81.24   | 94.8  | 92.99 | 16.77 |
> | Janus-Pro-1B  | 40.55  | 41.44  | 26.1   | 26.66 | 92.39  | 96.42 | 88.78 | 32.3  |
> | Lumina-Image-2.0 (2B) | 43.74  | 43.79  | 29.2   | 34.28  | 86.86      | 93.47 | 93.81 | 39.68 |
> | BAGEL-7B-MoT  | 47.04  | 46.35  | 33.8   | 39.59 | 97.41  | 97.99 | 94.68 | 49.43 |
>
> Our analysis of these results reveals several insights into what drives performance on detail-rich, long-prompt generation:
>
> - **Janus-1.3B**: The original Janus model establishes a solid baseline. Its performance validates *its core design of decoupling the understanding and generation encoders, which helps mitigate task conflict*. Additionally, its *unified architecture inherently enables it to comprehend longer prompts*. However, its capabilities **are constrained by its 1.3B scale and initial training data**, leading to moderate scores in complex dimensions like Character Locations and Entity Relationships.
> - **JanusFlow-1.3B**: While JanusFlow achieves a very high Style score (suggesting high-quality image aesthetics), it underperforms the original Janus in most attribute-binding and spatial tasks. This result suggests that while the flow mechanism can improve overall image quality, **it may compromise the model's adherence to the fine-grained compositional details** prevalent in DetailMaster's prompts.
> - **Janus-Pro-1B**: Janus-Pro demonstrates a significant improvement across all metrics compared to its predecessors. As discussed in its paper, the "Pro" version **benefits from an optimized training strategy and substantially expanded training data**. This directly enhances its ability to interpret and render the complex compositional requirements of our benchmark, showcasing the critical role of data quality and training refinement in handling long prompts.
> - **Lumina-Image-2.0**: Lumina-Image 2.0 continues this upward trend. Its strong performance, especially the notable jump in Character Locations and Entity Relationships, can be attributed to its two key innovations mentioned in their work: **the Unified Next-DiT architecture and the Unified Captioner (UniCap)**. UniCap is specifically designed to produce high-quality, multi-granularity textual descriptions for T2I tasks. **Training on this highly descriptive data** aligns perfectly with the demands of our DetailMaster benchmark, enabling the model to better ground visual elements to detailed textual specifications.
> - **BAGEL-7B-MoT**: BAGEL achieves the highest performance across all dimensions. Its success stems from a combination of factors discussed in its paper: (1) **Model Scale:** At 7B parameters, it has a significantly larger capacity for knowledge and reasoning. (2) **MoT Architecture:** It employs a Mixture-of-Transformers (MoT) architecture with distinct experts for understanding and generation. This design minimizes task interference more effectively than a simple encoder decoupling, allowing each expert to specialize. (3) **Large-scale Interleaved Data:** Its training on trillions of tokens of interleaved multi-modal data equips it with superior compositional reasoning.
>
> In summary, these results demonstrate that performance on DetailMaster is strongly correlated with *architectural choices that mitigate task conflict* (e.g., decoupled encoders, MoT), *the richness of the training data* (e.g., UniCap), *and overall model scale*. This analysis further validates **the utility of our benchmark in discerning the key capabilities and limitations** of advanced unified models. **We have newly added this discussion to the appendix of our paper (Appendix P)**.
>
> Furthermore, **our revised paper includes a new analysis of unified models**: while overall accuracy still drops with prompt length, their attribute alignment for generated characters is more robust (**Figure 7 & 8**). We attribute this enhanced stability to their core design principles, which *combine architectural specialization* (e.g., MoT, decoupled encoders) *with data-centric training on rich, descriptive captions*. This result validates that the unified model paradigm is a significant step forward and underscores our benchmark's utility in quantifying these vital, compositional advancements.
>
> (To be continued.)

---

> ### Author Response · Authors · 2025-11-20
> **Responses to Reviewer YsLh [Part 6/7]**
>
> (Continued from the previous page. W1.4)
>
> # W1.5: A Superior LLM/MLLM Encoder Yields Enhancements in Generative Output
>
> Thank you for this insightful and forward-looking question. We agree completely that the recent trend of using powerful LLM/MLLMs as text encoders represents a significant advancement in T2I generation, particularly for enhancing text understanding. It is meaningful and valuable to evaluate these advanced models on our benchmark to gauge their progress in handling long, detail-intensive prompts.
>
> In response to this excellent suggestion, we conduct a new set of experiments on two representative models that leverage LLM-based or MLLM-based encoders: **SANA**, which employs a Gemma2 as its encoder, and **Qwen-Image**, which utilizes Qwen2.5-VL. The evaluation results on our benchmark are summarized below:
>
> | Model | Object | Animal | Person | Character Locations | Background | Light | Style | Entity Relationships |
> | - | - | - | - | - | - | - | - | - |
> | SANA | 40.79 | 38.88 | 24.74 | 22.91 | 89.8 | 94.51 | 73.34 | 29.56 |
> | Qwen-Image | 51.01 | 49.11 | 39.3 | 46.98 | 98.35 | 98.98 | 96.08 | 60.04 |
>
> To provide a more comprehensive analysis, we compare these new results against the existing models evaluated in our paper, especially the previous SOTA models like GPT Image-1 and FLUX.1-dev:
>
> 1. **Impact of Advanced Encoders:** Both SANA and Qwen-Image substantially outperform earlier open-source models that rely on CLIP or T5 encoders. For example, SANA's performance on "Entity Relationships" is more than double that of DeepFloyd IF. This directly confirms the reviewer's hypothesis: **a stronger text encoder is crucial for parsing the complex compositional and relational details present in long prompts**.
> 2. **Analysis of SANA:** SANA shows a solid improvement over older architectures, particularly in scene-level attributes. However, its performance on fine-grained compositional tasks like "Character Attributes", "Character Locations," and "Entity Relationships" still lags behind top-tier models. This suggests that *while a better encoder provides a stronger semantic foundation, other architectural components of the diffusion model* (e.g., SANA's focus on efficiency with linear attention) *and the training data composition remain critical factors in achieving the highest level of detail adherence*.
> 3. **Analysis of Qwen-Image:** Qwen-Image's performance is particularly impressive, achieving results that are highly competitive with, and in some areas even surpass, the leading proprietary model, GPT Image-1.
>
> - - **Superior Compositionality:** Qwen-Image achieves an "Entity Relationships" score of 60.04%, which is very close to GPT Image-1's 63.07% and significantly higher than FLUX.1-dev's 47.49%. This highlights its exceptional ability to understand and render complex spatial and interactive relationships between multiple characters.
>   - **Strong Attribute Binding:** Its scores on "Character Attributes" (e.g., 49.11% for Animal, 39.30% for Person) are on par with or exceed those of previous SOTA models, indicating robust attribute binding even under high detail loads.
>   - **SOTA-Level Performance:** Qwen-Image's strong performance across all four dimensions validates that the combination of a powerful MLLM encoder (Qwen2.5-VL) and a robust diffusion backbone is a highly effective path toward mastering long-prompt generation. Nevertheless, its scores remain far from perfect, indicating considerable room for improvement.
>
> These new experiments **also underscore the unique contribution of our benchmark**. Simpler benchmarks focusing on short prompts might not reveal such a clear performance gap between models with different encoders. However, DetailMaster, with its average prompt length of over 284 tokens and its fine-grained evaluation across four critical dimensions, **provides the necessary complexity and granularity** to quantify the benefits of advanced encoders and identify persistent challenges. It highlights that even with superior text understanding, the accuracy on the most difficult compositional dimensions (Character Locations and Entity Relationships) remains far from perfect. In addition, handling intricate, long-form instructions remains an unsolved problem.
>
> We are grateful for the reviewer's suggestion, which helps us further strengthen our analysis. **We have newly added these new results and the accompanying discussion to the revised version of our manuscript (Appendix O)**.
>
> Furthermore, **we have updated our paper with a new analysis plotting the performance of these two models against prompt length (Figure 7 & 8)**. While Figure 7 indicates that even these models experience performance degradation on longer prompts, Figure 8 reveals *an improvement in attribute alignment for successfully generated characters*. This validates that using better LLM/MLLM encoders are a promising direction for enhancing compositional fidelity in long-prompt scenarios.
>
> (To be continued.)

---

> ### Author Response · Authors · 2025-11-20
> **Responses to Reviewer YsLh [Part 7/7]**
>
> (Continued from the previous page. W1.5)
>
> # Q: Qwen2.5-VL-3B-Instruct as a Viable Evaluator.
>
> Thank you for this excellent suggestion. We agree with the reviewer that the computational requirements of our MLLM-based evaluation, particularly the VRAM needed for Qwen2.5-VL-7B-Instruct, could present a barrier for some research teams. **Introducing a more lightweight evaluator is a crucial step toward increasing the accessibility and widespread adoption of our benchmark.**
>
> Fortunately, our evaluation protocol's robustness is not solely dependent on the MLLM. As detailed in our paper, *we integrate a suite of auxiliary techniques* (such as open-set object detectors and structured verification steps) that ground the evaluation in objective features and enhance its accuracy. This design significantly *mitigates the risk of performance degradation when switching to a smaller evaluator model*.
>
> To empirically validate this and directly address your concern, **we conduct a new set of experiments, re-running our entire evaluation pipeline using a smaller evaluator,** **Qwen2.5-VL-3B-Instruct**. The comprehensive results are presented in the table below.
>
>
>
> | Model | Object | Animal | Person | Character Locations | Background | Light  | Style  | Entity Relationships |
> | - | - | - | - | - | - | - | - | - |
> | SD1.5 | 20.23 | 28.76 | 13.39 | 11.62 | 32.24 | 80.99 | 80.57 | 10.00 |
> | SD-XL  | 24.08 | 29.16 | 17.91 | 15.51 | 36.48 | 83.18 | 69.09 | 15.85 |
> | DeepFloyd IF | 29.57 | 39.39 | 27.22 | 19.70  | 34.20 | 83.36 | 85.71 | 17.73 |
> | SD3.5 Large | 45.25 | 45.85 | 33.82 | 40.04 | 93.62 | 96.38 | 94.99 | 51.85 |
> | FLUX.1-dev | 47.61 | 44.39 | 35.25 | 45.20 | 96.83 | 98.81 | 93.95 | 57.25 |
> | Gemini 2.0 Flash | 48.88 | 46.75 | 34.33 | 49.98 | 97.53 | 98.34 | 96.32 | 59.82 |
> | GPT Image-1 | 54.82 | 46.16 | 40.98 | 55.47 | 98.52 | 99.24 | 95.75 | 69.63 |
> | LLM4GEN  | 21.35 | 29.70 | 18.54 | 12.63 | 39.28 | 82.67 | 58.32 | 11.79 |
> | LLM Blueprint | 20.34 | 26.16 | 14.73 | 23.82 | 65.61 | 89.95 | 66.62 | 20.13 |
> | ELLA | 30.53 | 34.60 | 22.08 | 20.26 | 56.51 | 88.38 | 41.15 | 23.62 |
> | LongAlign | 28.86 | 33.14 | 15.62 | 19.96 | 85.43 | 94.97 | 82.09 | 27.71  |
> | ParaDiffusion | 33.65 | 34.43 | 24.40 | 25.73 | 89.38 | 96.35 | 65.87 | 36.12 |
>
> As the results demonstrate, while the absolute scores differ slightly from the original evaluation, **the comparative rankings among the models and the general score trends remain unchanged**. For **general-purpose models**, more advanced models consistently achieve higher scores. Similarly, **long-prompt optimized models** continue to outperform their baselines (SD1.5 and SD-XL).
>
> Regarding more specific details, we observe consistent trends that align with our original findings:
>
> - A clear performance chasm persists among general-purpose models. A significant gap separates older models (SD1.5, SD-XL, DeepFloyd IF) from advanced ones (SD3.5 Large, FLUX.1-dev, etc.).
> - ParaDiffusion maintains its superior performance in "Character Locations" and "Entity Relationships" compared to other long-prompt optimized models.
> - While top-tier general-purpose models achieve near-perfect scores, the long-prompt optimized models show varied success. For instance, LongAlign and ParaDiffusion demonstrate remarkable gains in Background and Light fidelity, whereas others like ELLA show more modest improvements, especially in the Style category.
> - We also compute **the Kendall's Tau Correlation Coefficient** between the evaluation results obtained with the two MLLMs. The correlation coefficients for each metric are as follows: 0.9697, 0.9090, 1.0, 1.0, 0.9394, 0.9394, 0.9090, and 0.9394. These consistently high values (all approaching 1.0, achieving an average value surpassing 0.95) reflect **a strong positive correlation** between the two MLLM evaluators, confirming the consistent scoring trends across the evaluated models.
>
> This high correlation in evaluation outcomes confirms that **Qwen2.5-VL-3B-Instruct is a viable and resource-efficient alternative** for our benchmark, preserving the integrity of the relative model comparisons.
>
> To enhance the accessibility of DetailMaster, **we have newly included these results and the lightweight evaluation protocol in Appendix K of our revised manuscript**. This will provide researchers with a resource-efficient option for model assessment and comparison. Thank you again for the constructive feedback!
>
> ---
>
> # Closing Remarks
>
> We would like to express our sincere gratitude for your suggestions and queries. As per your suggestions, we invest considerable time and effort into conducting additional experiments. We hope that our responses have addressed your concern and that you will consider re-evaluating our paper. If you have any further questions or suggestions, please do not hesitate to reach out. Thank you once again for your time and effort.

---

> ### Author Response · Authors · 2025-11-28
>
> Dear Reviewer YsLh,
>
> Thank you again for your exceptionally detailed and thought-provoking review of our paper. Your questions push us to delve much deeper into the analysis, and we are very grateful for that.
>
> As the discussion period nears its end, we would be very appreciative if you could let us know whether our responses and new experiments have addressed your points. We specifically hope that the new controlled ablation study has clarified the underlying reasons for diverse model behaviors, and that the added evaluations on unified and LLM-encoder models have provided the broader context you are looking for.
>
> If you have any further thoughts, we would welcome the opportunity to discuss it.
>
> Thank you for your invaluable time and engagement with our work.
>
> Best, authors

---

### Official Review · Reviewer_PbGa · 2025-10-28

**Soundness:** 3
**Presentation:** 2
**Contribution:** 3
**Rating:** 6
**Confidence:** 4

**Summary:**

This paper proposes the comprehensive benchmark to address extended textual inputs that contain complex compositional requirements. The benchmark introduces four critical evaluation dimensions: Character Attributes, Structured Character Locations, Multi-Dimensional Scene Attributes, and Spatial/Interactive Relationships.

**Strengths:**

1. The paper proposes the comprehensive compositional dataset on long, complex prompts.
2. The paper is well-written and easy-to-follow.
3. The experiments are extensive.

**Weaknesses:**

1. The paper lacks discussion of ConceptMix, which targets at compositional T2I generation.
2. The attribute pipeline relies on MLLM (e.g.,  use MLLM to identify its background composition, lighting conditions, and stylistic elements), which may introduce hallucinations or mistakes. And use MLLM as evaluators may still introduce problems though authors tried to mitigate. For example, the evaluation results are not easy to reproduce.

[A] Wu X, Yu D, Huang Y, et al. Conceptmix: A compositional image generation benchmark with controllable difficulty[J]. Advances in Neural Information Processing Systems, 2024, 37: 86004-86047.

**Questions:**

1. What are the ways to prevent hallucinations of MLLM in pipeline construction?

---

> ### Author Response · Authors · 2025-11-20
> **Responses to Reviewer PbGa [Part 1/2]**
>
> Dear Reviewer PbGa, many thanks for your appreciation of the comprehensiveness and soundness of our work! In our rebuttal, we provide point-by-point responses to your valuable comments, with all additional discussions incorporated into our revised paper. We sincerely hope our responses can fully address your concerns and strengthen your confidence in the acceptance of our paper.
>
> ---
>
> # W1: Inclusion of ConceptMix in Related Works
>
> We sincerely thank the reviewer for bringing the excellent work, ConceptMix, to our attention. This is a valuable suggestion that helps us better contextualize our work.
>
> ConceptMix proposes a scalable, controllable, and automated benchmark specifically designed to evaluate the compositional capabilities of T2I models. A key contribution of this work is the automatic generation of prompts by systematically combining a varying number of discrete visual concepts (controlled by a difficulty parameter k). The evaluation is also automated by leveraging a strong VLM to verify the presence of each specified concept in the generated image. This approach provides a precise way to measure how well models handle increasing combinatorial complexity.
>
> We agree that a discussion of ConceptMix is essential for a comprehensive overview of the T2I evaluation landscape. **We have now added a discussion of this work to our** **Related Works** **section** (**L131-132**). The new text helps to situate our DetailMaster benchmark by highlighting the complementary nature of our approaches: while ConceptMix focuses on the challenge of combining an increasing number of discrete concepts, our work addresses the distinct but related challenge of handling long, descriptively rich prompts with complex, nested details.
>
> ---
>
> # W2.1 & Q1: Clarification on the Permissibility of Hallucinations and Methods for Ensuring Prompt **Reasonableness**
>
> Thank you for this valuable feedback. We would like to clarify what appears to be a misunderstanding regarding the design of our data construction pipeline.
>
> ## **Human Evaluation Validates Rare Hallucination Occurrence**
>
> To address the concern regarding potential MLLM hallucinations in our data synthesis pipeline, we would like to highlight **the human expert evaluation** presented in Appendix G, which found that **93.6% of samples maintain visual fidelity with the source images**.
>
>
>
> ## **Clarification on the Evaluation's Sole Dependence on the Final Prompt**
>
> Moreover, it is important to emphasize that **the original image functions solely as** `an initial seed` **to inspire the generation of high-quality, detailed prompts**. The final prompt is not designed to faithfully replicate the content of the original image.
>
> **The core evaluation in our benchmark is exclusively aligned with the final prompt, not the original image.** When we assess model performance, we are measuring how well the generated images match the content described in the final prompt, regardless of whether that content originated from the original image.
>
>
>
> ## **Verification Step for Prompt Reasonableness**
>
> **To address concerns regarding prompt reasonableness, we have incorporated multiple verification mechanisms in our original paper.** After the initial attribute extraction for the main character, we re-deploy the MLLM to assess whether the extracted features are indeed coherent and characteristic of that character. Additionally, our framework complements the MLLM with an open-set object detection model to verify and refine character locations. These processes help **ensure that the final prompt is not only richly detailed but also semantically reasonable and feasible for image generation**.
>
>
>
> We hope this explanation fully resolves your concern. To prevent similar misunderstandings, **we have newly added relevant explanations in our paper to better elucidate our data synthesis pipeline (L214-215)**. In addition, **we have newly revised Figure 2** (pipeline diagram) to explicitly label the input image as "Semantic Seed".
>
>
> (To be continued.)

---

> ### Author Response · Authors · 2025-11-20
> **Responses to Reviewer PbGa [Part 2/2]**
>
> (Continued from the previous page. W2.1 & Q1)
>
> # W2.2 Robustness and Reproducibility of Our Evaluation with MLLM
>
> Thank you for raising this important point regarding the potential reproducibility issues when using MLLMs as evaluators. We appreciate your concern and would like to clarify a possible misunderstanding.
>
> In our appendix, we have thoroughly discussed and demonstrated that employing MLLMs for evaluation is robust and does not compromise reproducibility. Specifically, in **Appendix I of our original paper**, we *replaced the Qwen2.5-VL-7B-Instruct evaluator with InternVL3-9B* and, during the rebuttal phase, also experimented with *Qwen2.5-VL-3B-Instruct*. In both cases, we observed that **the comparative rankings among the models and the general score trends remain unchanged.** For general-purpose T2I models, more advanced models consistently achieve higher scores. Similarly, long-prompt optimized T2I models outperform the baseline models SD1.5 and SD-XL. Moreover, the performance differences between models are consistently preserved. This cross-evaluator consistency provides strong evidence that *our benchmark's conclusions are robust*, and the extensive auxiliary techniques we introduced effectively mitigate potential biases introduced by MLLMs.
>
> Additionally, in **Appendix J of our original paper**, we modified the random seed used in the LLM-based evaluation and found that the scores for all T2I models exhibited **negligible variations (∆ < 0.5%)**. This robust consistency strongly validates the stability and reliability of our evaluation protocol, demonstrating that our auxiliary techniques effectively reduce the impact of inherent LLM randomization. Consequently, our approach ensures reproducible and dependable model comparisons, even in the presence of stochastic factors in LLM-based assessment.
>
> We hope these clarifications alleviate your concerns regarding reproducibility and reinforce the reliability of our evaluation pipeline.
>
> ---
>
> # Closing Remarks
>
> Thanks again for your comments! We hope that our responses have addressed your concern and can convince you to lean more toward acceptance of our paper. Kindly do not hesitate to communicate with us if you have any further inquiries or recommendations. We appreciate your time and feedback immensely.

---

> ### Author Response · Authors · 2025-11-28
>
> Dear Reviewer PbGa,
>
> We would like to extend our sincere thanks for your valuable feedback on our paper.
>
> With the author-reviewer discussion period ending soon, we want to kindly follow up and ask if our response has clarified the points you raised, particularly regarding our data generation process and the robustness of our MLLM-based evaluator. We hope the detailed appendices on cross-evaluator consistency have helped resolve these concerns.
>
> Please let us know if anything in our rebuttal requires further explanation.
>
> Thank you again for your time and feedback.
>
> Best, authors

---

### Official Review · Reviewer_nKZ7 · 2025-10-29

**Soundness:** 2
**Presentation:** 2
**Contribution:** 2
**Rating:** 4
**Confidence:** 4

**Summary:**

This paper proposes DETAILMASTER for evaluating the consistency between images and text generated by T2I models under complex long-text conditions. DETAILMASTER starts from open-source image-caption datasets with dense annotations, further expanding on Character Attributes, Structured Character Locations, Spatial/Interactive Relationships, and Multi-Dimensional Scene Attributes in the prompt through MLLM & LLM, synthesizing the final version of DETAILMASTER. The paper evaluates several general T2I models as well as T2I models specifically optimized for long text, demonstrating that even the most advanced T2I models still need further improvement in aligning images and text under complex long-text prompts.

**Strengths:**

1. The consistency of text and images in complex long prompts is crucial for evaluating the capabilities of T2I models, and existing benchmarks are indeed lacking in this aspect;
2. The benchmark synthesis process in the paper comprehensively considers various aspects under long text prompts, such as Character Attributes, Structured Character Locations, and Multi-Dimensional Scene Attributes;
3. The paper conducts extensive experiments on existing open-source and closed-source models, indicating that the current state-of-the-art models still need further improvement in handling complex long texts.

**Weaknesses:**

1. Recent diffusion models that use MLLM as a text encoder, such as Hunayuan Image 3.0 and Qwen-Image, possess stronger text understanding capabilities. How do these models perform on DetailMaster?
2. During evaluation, DetailMaster needs to detect the bounding box for each character based on the Character List. How does it handle cases when the prompt contains multiple repeated characters and there are interactions between these repeated characters?
3. Due to the inherent hallucinations of LLMs/MLLMs, there might be inconsistencies between the final prompt and the image content in DetailMaster's benchmark creation process. The paper does not develop a secondary verification process to ensure higher accuracy.
4. In section 3.2.3, the paper admits that using a single MLLM family (i.e., QwenVL) for both data curation and evaluation could raise concerns about potential self-enhancement bias. Although evaluation results on previous t2i models using InternVL and QwenVL indicate consistent relative rankings, I am still curious whether the evaluation results of diffusion models like Qwen-Image, which use QwenVL as a text encoder, are consistent between QwenVL and InternVL evaluations in DetailMaster.

**Questions:**

See Weaknesses

---

> ### Author Response · Authors · 2025-11-20
> **Responses to Reviewer nKZ7 [Part 1/4]**
>
> Dear Reviewer nKZ7, we sincerely appreciate your positive recognition of the comprehensive coverage and the extensive experiments in our work! In response to your comments, we provide detailed discussions along with additional experiments below, all of which have been integrated into our revised paper (blue text). We hope the responses will convince you to lean more toward the acceptance of our paper.
>
> ---
>
> # W1: The Performance of Recent LLM-Based/MLLM-Based Diffusion Models on DetailMaster
>
> Thank you for this insightful and forward-looking question. We agree completely that the recent trend of using powerful LLM/MLLMs as text encoders represents a significant advancement in T2I generation, particularly for enhancing text understanding. It is meaningful and valuable to evaluate these advanced models on our benchmark to gauge their progress in handling long, detail-intensive prompts.
>
> In response to this excellent suggestion, we conduct a new set of experiments on two representative models that leverage LLM-based or MLLM-based encoders: **SANA**, which employs a Gemma2 as its encoder, and **Qwen-Image**, which utilizes Qwen2.5-VL. The evaluation results on our benchmark are summarized in the table below:
>
> | Model      | Object | Animal | Person | Character Locations | Background | Light | Style | Entity Relationships |
> | ---------- | ------ | ------ | ------ | ------------------- | ---------- | ----- | ----- | -------------------- |
> | SANA       | 40.79  | 38.88  | 24.74  | 22.91               | 89.8       | 94.51 | 73.34 | 29.56                |
> | Qwen-Image | 51.01  | 49.11  | 39.3   | 46.98               | 98.35      | 98.98 | 96.08 | 60.04                |
>
> To provide a more comprehensive analysis, we compare these new results against the existing models evaluated in our paper, especially the previous SOTA models like GPT Image-1 and FLUX.1-dev:
>
> 1. **Impact of Advanced Encoders:** Both SANA and Qwen-Image substantially outperform earlier open-source models that rely on CLIP or T5 encoders. For example, SANA's performance on "Entity Relationships" is more than double that of DeepFloyd IF. This directly confirms the reviewer's hypothesis: **a stronger text encoder is crucial for parsing the complex compositional and relational details present in long prompts**.
> 2. **Analysis of SANA:** SANA shows a solid improvement over older architectures, particularly in scene-level attributes. However, its performance on fine-grained compositional tasks like "Character Attributes", "Character Locations," and "Entity Relationships" still lags behind top-tier models. This suggests that *while a better encoder provides a stronger semantic foundation, other architectural components of the diffusion model* (e.g., SANA's focus on efficiency with linear attention) *and the training data composition remain critical factors in achieving the highest level of detail adherence.*
> 3. **Analysis of Qwen-Image:** Qwen-Image's performance is particularly impressive, achieving results that are highly competitive with, and in some areas even surpass, the leading proprietary model, GPT Image-1.
>
> - - **Superior Compositionality:** Qwen-Image achieves an "Entity Relationships" score of 60.04%, which is very close to GPT Image-1's 63.07% and significantly higher than FLUX.1-dev's 47.49%. This highlights its exceptional ability to understand and render complex spatial and interactive relationships between multiple characters.
>   - **Strong Attribute Binding:** Its scores on "Character Attributes" (e.g., 49.11% for Animal, 39.30% for Person) are on par with or exceed those of previous SOTA models, indicating robust attribute binding even under high detail loads.
>   - **SOTA-Level Performance:** Qwen-Image's strong performance across all four dimensions validates that the *combination* of a powerful MLLM encoder (Qwen2.5-VL) and a robust diffusion backbone is a highly effective path toward mastering long-prompt generation. Nevertheless, **its scores remain far from perfect**, indicating considerable room for improvement.
>
> (To be continued.)

---

> ### Author Response · Authors · 2025-11-20
> **Responses to Reviewer nKZ7 [Part 2/4]**
>
> (Continued from the previous page. W1)
>
> These new experiments **also underscore the unique contribution of our benchmark**. Simpler benchmarks focusing on short prompts might not reveal such a clear performance gap between models with different encoders. However, DetailMaster, with its average prompt length of over 284 tokens and its fine-grained evaluation across four critical dimensions, **provides the necessary complexity and granularity** to quantify the benefits of advanced encoders and identify persistent challenges. The results highlight that *even with superior text understanding, the accuracy on the most difficult compositional dimensions (Character Locations and Entity Relationships) remains far from perfect*. In addition, handling intricate, long-form instructions remains an unsolved problem.
>
>
>
> We are grateful for the reviewer's suggestion, which has helped us further strengthen our analysis. **We have newly added these new results and the accompanying discussion to the revised version of our manuscript (Appendix O)**.
>
> Furthermore, **we have updated our paper with a new analysis plotting the performance of these two models against prompt length (Figure 7, Figure 8)**. While Figure 7 indicates that even these models experience performance degradation on longer prompts, Figure 8 reveals an improvement in attribute alignment for successfully generated characters. This validates that using better LLM-based or MLLM-based encoders are a promising direction for enhancing compositional fidelity in long-prompt scenarios.
>
>
> ---
>
>
> # W2: Robust Disambiguation of Interactive Repeated Characters
>
> Thank you for raising this important question regarding the handling of repeated characters. Nevertheless, such cases are rare within our benchmark, and the reasons are explained below.
>
> ## **Regarding Low Character Repetition via Attributive Modifiers**
>
> In our data construction process, the Character List is constructed with **characters described using representative modifiers**, which inherently prevents the occurrence of repeated characters. Specifically, the names of the main characters are extracted from detailed captions in the data sources (DOCCI and Localized Narratives) using an MLLM (**the MLLM also captures initial modifiers for each character**).  Since the original detailed captions are manually annotated, the annotators intentionally distinguish each main character by assigning different modifiers. As a result, the modifiers associated with each character in the Character List are distinct, enabling clear differentiation even among characters that might otherwise share the same base name.
>
>
>
> ## **Statistics on the Infrequency of Repeated Characters**
>
> To further address your concern, we conduct a new statistical analysis across all samples within DetailMaster. The results indicate that **in 97.49% of the samples, no repeated characters are present in the Character List**, as different characters can be clearly distinguished based on their respective modifiers. We have newly incorporated this statistical analysis into the revised version of our paper (**L1099-1103**).
>
>
>
> ## **Robust Localization Method**
>
> Moreover, thanks to the integration of the MLLM and the open-set object detection model, *our detection approach is capable of accurately locating characters even when complex interactions occur between them*. This is achieved by leveraging the representative modifiers associated with each character to instruct the two models, ensuring that the bounding boxes are correctly identified and assigned.
>
> We appreciate your insightful comment and hope this clarification adequately addresses your question.
>
>
> (To be continued.)

---

> ### Author Response · Authors · 2025-11-20
> **Responses to Reviewer nKZ7 [Part 3/4]**
>
> (Continued from the previous page. W2)
>
> # W3: Clarification on the Role of Original Images and Prompt Alignment
>
> Thank you for raising this important point. However, there is a misunderstanding regarding the core design principle of our benchmark.
>
> ## **Human Evaluation Validates Rare Hallucination Occurrence**
>
> To address the concern regarding potential MLLM hallucinations in our data synthesis pipeline, we would like to highlight **the human expert evaluation** presented in Appendix G, which found that **93.6% of samples maintain visual fidelity with the source images**.
>
>
>
> ## **Clarification on the Evaluation's Sole Dependence on the Final Prompt**
>
> Moreover, we would like to emphasize that **the original image serves merely as** `an initial seed` **to inspire the generation of high-quality, detailed prompts.** The final prompt generated through the construction process **is not intended to be a faithful description** of the original image content.
>
> **The core evaluation in our benchmark is exclusively aligned with the final prompt, not the original image.** When we assess model performance, we are measuring how well the generated images match the content described in the final prompt, regardless of whether that content originated from the original image.
>
> ## **Verification Step for Prompt Reasonableness**
>
> Furthermore, to address the underlying concern about prompt reasonableness, **we have already implemented an additional verification step in our original paper**. After the initial attribute extraction for the main character, we re-deploy the MLLM to assess whether the extracted features are indeed coherent and characteristic of that character. This process helps ensure that the final prompt is not only richly detailed but also semantically reasonable and feasible for image generation.
>
> Consequently, we do not require an additional verification process to ensure alignment with the original image. **All annotations and evaluations in our benchmark are designed to capture adherence to the final prompt**, including any content that might be considered "hallucinated."
>
> We hope our response fully addresses your concern. To prevent similar misunderstandings, **we have newly added relevant explanations in the paper to better elucidate our data synthesis pipeline (L214-215)**. In addition, **we have newly revised Figure 2** (pipeline diagram) to explicitly label the input image as "Semantic Seed".
>
>
> (To be continued.)

---

> ### Author Response · Authors · 2025-11-20
> **Responses to Reviewer nKZ7 [Part 4/4]**
>
> (Continued from the previous page. W3)
>
> # W4: Robust Performance of Qwen-Image Under Both QwenVL and InternVL Assessments
>
> Thank you for raising this important point. The concern about potential self-enhancement bias, especially for models like Qwen-Image that share an architectural family with our primary MLLM evaluator (QwenVL), is a valid and crucial aspect of benchmark robustness.
>
> To directly address this, we conduct **a new evaluation of Qwen-Image on our benchmark** using both the original **Qwen2.5-VL-7B-Instruct** evaluator and an alternative, architecturally distinct evaluator, **InternVL3-9B**. The results are presented in the table below:
>
> | Model                  | Object | Animal | Person | Character Locations | Background | Light | Style | Entity Relationships |
> | ---------------------- | ------ | ------ | ------ | ------------------- | ---------- | ----- | ----- | -------------------- |
> | Qwen-Image w/ QwenVL   | 51.01  | 49.11  | 39.3   | 46.98               | 98.35      | 98.98 | 96.08 | 60.04                |
> | Qwen-Image w/ InternVL | 69.64  | 84.5   | 62.17  | 77.54               | 99.56      | 99.66 | 96.45 | 86.55                |
>
> As the table demonstrates, Qwen-Image's absolute scores are higher when assessed by the InternVL3-9B evaluator. This observation is consistent with our findings in Appendix I (Table 9), where we noted that InternVL3-9B exhibits a more permissive evaluation tendency, leading to a general score inflation across all tested models.
>
> However, the more critical finding is the consistency of the model's relative performance. **Qwen-Image's performance trend compared to other evaluated models remains unchanged.** Specifically:
>
> - **Preservation of Relative Ranking:** Qwen-Image's top-tier ranking relative to other diffusion models (such as GPT Image-1 and FLUX.1-dev) is preserved. It continues to demonstrate excellent performance, regardless of whether the evaluator is from the Qwen family or not.
> - **Refuting Self-Enhancement Bias:** If a significant self-enhancement bias were present, we would expect Qwen-Image's performance advantage to diminish or disappear when evaluated by a non-Qwen MLLM. Our results show the opposite: **Qwen-Image's strong performance is consistently recognized by an independent evaluator.** This indicates that its high scores are attributable to its genuine compositional generation capabilities rather than an artifact of the evaluation setup.
>
> In summary, this new experiment provides direct evidence that our evaluation framework is robust and that **Qwen-Image's strong performance on DetailMaster is not a result of self-enhancement bias.** We have newly added these results to the appendix to further strengthen our paper's claims of robustness (**Appendix J**).
>
> ---
>
> # Closing Remarks
>
> We sincerely appreciate the valuable time you have spent in reviewing our paper.  We hope that our responses have addressed your concern and that you will consider re-evaluating our paper. Please do not hesitate to contact us if you have any further inquiries or recommendations. Thank you!

---

> ### Author Response · Authors · 2025-11-28
>
> Dear Reviewer nKZ7,
>
> Thank you once again for your insightful and forward-looking review of our paper. Your comments about recent LLM-based/MLLM-based models and the robustness of our evaluation are particularly valuable.
>
> As the discussion period is drawing to a close, we would be very grateful to know if our rebuttal and the new experiments—specifically the evaluation of models like Qwen-Image and SANA, and the cross-validation using an independent evaluator (InternVL3-9B) to check for bias—have successfully addressed the concerns you raised.
>
> If you have any further questions, please do not hesitate to let us know.
>
> Thank you for your time and guidance.
>
> Best, authors

---

### Official Review · Reviewer_5wZt · 2025-10-30

**Soundness:** 3
**Presentation:** 3
**Contribution:** 2
**Rating:** 4
**Confidence:** 5

**Summary:**

The paper introduces a benchmark for assessing T2I models on their ability to faithfully handle long prompts. To craft the benchmark, the authors follow a data curation pipeline for the evaluation dataset. The paper also includes analyses and insights into the limitations of current diffusion models w.r.t handling long prompts.

**Strengths:**

* Data curation pipeline
* Analysis of limitations of current benchmarks pertaining to long prompts
* Robustness and validity of the benchmark through human evaluation

**Weaknesses:**

* Lack of controlled experiments to drive the insights and analyses in Section 4. It would have been much better to take a single model architecture and ablate it under different setups to drive the insights. More on this in "Questions".
* Lack of results with several models that are known for handling long and complex prompts (such as SANA [1], Lumina-Next [2], and QwenImage [3]).
* It's said in the paper multiple times (L39, for example) that T2I models are trained on short-length prompts. However, many recent models actually leverage denser prompts during their training phase (SANA, for example).

**References**

[1] SANA: Efficient High-Resolution Image Synthesis with Linear Diffusion Transformers; Xie et al.; 2024.

[2] Lumina-Image 2.0: A Unified and Efficient Image Generative Framework; Qin et al.; 2025.

[3] Qwen-Image Technical Report; Wu et al.; 2025.

**Questions:**

> Our analysis reveals fundamental limitations in compositional reasoning, demonstrating that current encoders flatten complex grammatical structures and that diffusion models suffer from attribute leakage under detail-intensive conditions.

Could the authors reference the sections where this was analyzed? Section 4 didn't seem to address these points.

> To achieve higher precision and greater detail, we subsequently reprocess the corresponding 4,565 samples using Qwen2.5-VL-72B-Instruct as both the LLM and the MLLM. This new process generates an improved dataset containing 4,116 prompts with an average token length of 284.89.

How is the number of prompts changing from 4,565 to 4,116? How were they validated? It would also be beneficial to the community if the authors could include all the system prompts that were used throughout this work.

> Section 4 analyses

It would have been better and more helpful to fine-tune a specific architecture on image and long prompt pairs, and then study the effects. Models like QwenImage already leverage an LLM backbone for encoding the text prompts and were also trained with a longer sequence length. So, I believe this is feasible. The models mentioned in "Long prompt training matters more than increasing token capacity." section all have varying amounts of confounding factors and hence it makes it difficult to drive educated observations.

> Datasets

It's unclear what proportions of the samples from each dataset were used to construct the benchmark. It could also be beneficial to include some commentary about how the original captions aren't long enough and miss the important, desirable details (possible through some quantifiable numbers). For example, what character attributes are missing in the original captions?

---

> ### Author Response · Authors · 2025-11-20
> **Responses to Reviewer 5wZt [Part 1/6]**
>
> Dear Reviewer 5wZt, we sincerely appreciate the time and effort you have dedicated to reviewing our paper, as well as the valuable feedback and suggestions you provide! Following your constructive suggestions, we conduct several additional experiments, which require considerable time and effort. Our revised paper now includes these additional experiments and analysis, with the new text marked in blue. Below, we provide detailed responses to each of your comments, with the hope that this will encourage you to lean toward the acceptance of our paper.
>
> ---
>
> # W1 & Q3: Ablation Study on the Token Constraint and Long Prompt Training Data
>
> We sincerely thank the reviewer for the insightful feedback and constructive suggestions. We agree that the initial analysis in Section 4, which compared various existing models, could be strengthened by `a more controlled experiment using a single model architecture` to isolate variables.
>
> ## **Clarification Regarding the Comparative Analysis in Our Original Paper**
>
> However, we would like to explain that the long-prompt optimized models introduced in our original paper were all intentionally selected because they are optimizations over the common backbones of SD-1.5 and SD-XL. While the potential confounding factors may exist, **the overall evaluation results are largely consistent with the relative merits of the different optimization schemes**, thus providing a valuable high-level assessment of their effectiveness.
>
> ## **Setup for New Experiments**
>
> In response to your request for a more controlled experiment on a single architecture, **we perform a new ablation on SD-XL that disentangles the effects of (i) increasing the token limit and (ii) explicitly training on long prompts.**
>
> We modify SD-XL by integrating a T5 text encoder alongside the original CLIP encoders. To fuse the features from the T5 encoder, we adopt a Timestep-Aware Semantic Connector, similar to the mechanism used in ELLA. This structure allows the model to leverage the extended prompt limit and rich semantic representation of T5 *without discarding the capabilities learned from CLIP features*. For efficiency, we keep the weights of the original SD-XL UNet, CLIP, and T5 frozen, *only training the new semantic connector*. As for the training data, we curate a dataset of 2M image-detailed caption pairs from the DOCCI dataset and the Localized Narratives dataset.
>
> We design four settings to evaluate the impact of *prompt limit and training prompt length*:
>
> 1. **77 Token Limit + Short-Prompt Training**.
> 2. **512 Token Limit + Short-Prompt Training**.
> 3. **77 Token Limit + Long-Prompt Training**.
> 4. **512 Token Limit + Long-Prompt Training**.
>
> Herein, `the short prompts` are generated by using Qwen2.5-VL-7B-Instruct to compress the original detailed captions to a length of 30 tokens or less.
>
> ## **Results and Analysis**
>
> We evaluate all four fine-tuned models on our benchmark. The results are as follows:
>
> | Model | Object | Animal | Person | Character Locations | Background | Light | Style | Entity Relationships |
> | - | - | - | - | - | - | - | - | - |
> | 77 Limit w/ Short-Prompt-Training | 24.38 | 28.17 | 16.42  | 9.39 | 29.31 | 74.72 | 46.64 | 8.43 |
> | 512 Limit w/ Short-Prompt-Training | 25.37 | 30.11 | 18.27  | 10.45 | 34.8 | 74.16 | 66.22 | 10.12 |
> | 77 Limit w/ Long-Prompt-Training | 27.44 | 31.54 | 18.32  | 10.93 | 33.7 | 76.62 | 53.72 | 10.66 |
> | 512 Limit w/ Long-Prompt-Training | 30.5 | 33.13 | 19.36  | 12.52 | 37.51 | 80.68 | 69.98 | 13.07 |
>
> These controlled experiments yield several key insights that strongly support the original conclusions of our paper:
>
> - **Both Factors Matter:** Comparing the results to the baseline (77 Limit w/ Short-Prompt), both expanding the prompt limit and using long-prompt training lead to performance gains across most compositional metrics, especially in attribute and location accuracy.
> - **Long-Prompt Training is More Impactful than Token Capacity Alone:** Comparison between the second and third setups reveals that the performance gain from training on long, descriptive prompts is more significant than the gain from merely increasing the prompt limit while training on short prompts. For instance, the model trained on long prompts with a 77-token limit outperforms the one trained on short prompts with a 512-token limit in Object, Animal, Person, Character Locations, and Entity Relationships.
> - **Synergistic Effect:** *The best performance* is achieved when both factors are combined (512 Limit w/ Long-Prompt Training). This result highlights a clear synergistic effect, suggesting that the combination of an expanded prompt limit and long-prompt training is a critical strategy for improving long-prompt adherence.
>
> We have newly added this ablation study to our revised paper to substantiate our claims more rigorously (**Section 4.5**). Thank you for pushing us to perform this valuable experiment, which has significantly strengthened our paper!
>
> (To be continued.)

---

> ### Author Response · Authors · 2025-11-20
> **Responses to Reviewer 5wZt [Part 2/6]**
>
> (Continued from the previous page. W1 & Q3)
>
> # W2: Additional Results with Models Specialized for Long Prompts
>
> Thank you for the valuable suggestion to include more recent models known for handling long prompts. In response, we conduct a new set of experiments, **evaluating a range of recently released models**, including:
>
> - `MLLM-encoder-based diffusion models:` SANA (Gemma2 encoder) and Qwen-Image (Qwen2.5-VL encoder);
> - `Advanced unified models:` Janus-1.3B, JanusFlow-1.3B, Janus-Pro-1B, Lumina-Image-2.0 (2B), and BAGEL-7B-MoT.
>
>
>
> ------
>
> ## **Evaluation on T2I Models with a Superior LLM/VLM Encoder**
>
> The recent trend of using powerful VLMs as text encoders represents a significant advancement in T2I generation, particularly for enhancing text understanding. We conduct a new set of experiments on two representative models that leverage MLLM-based encoders: **SANA**, which employs a Gemma2 as its encoder, and **Qwen-Image**, which utilizes Qwen2.5-VL. The evaluation results on our benchmark are summarized in the table below:
>
> | Model      | Object | Animal | Person | Character Locations | Background | Light | Style | Entity Relationships |
> | ---------- | ------ | ------ | ------ | ------------------- | ---------- | ----- | ----- | -------------------- |
> | SANA       | 40.79  | 38.88  | 24.74  | 22.91               | 89.8       | 94.51 | 73.34 | 29.56                |
> | Qwen-Image | 51.01  | 49.11  | 39.3   | 46.98               | 98.35      | 98.98 | 96.08 | 60.04                |
>
> To provide a more comprehensive analysis, we compare these new results against the existing models evaluated in our paper, especially the previous SOTA models like GPT Image-1 and FLUX.1-dev:
>
> 1. **Impact of Advanced Encoders:** Both SANA and Qwen-Image substantially outperform earlier open-source models that rely on CLIP or T5 encoders. For example, SANA's performance on "Entity Relationships" is more than double that of DeepFloyd IF. This directly confirms that **a stronger text encoder is crucial for parsing the complex compositional and relational details present in long prompts**.
> 2. **Analysis of SANA:** SANA shows a solid improvement over older architectures, particularly in scene-level attributes. However, its performance on fine-grained compositional tasks like "Character Attributes", "Character Locations," and "Entity Relationships" still lags behind top-tier models. This suggests that *while a better encoder provides a stronger semantic foundation, other architectural components of the diffusion model* (e.g., SANA's focus on efficiency with linear attention) *and the training data composition remain critical factors in achieving the highest level of detail adherence*.
> 3. **Analysis of Qwen-Image:** Qwen-Image's performance is particularly impressive, achieving results that are highly competitive with, and in some areas even surpass, the leading proprietary model, GPT Image-1.
>
> - - **Superior Compositionality:** Qwen-Image achieves an "Entity Relationships" score of 60.04%, which is very close to GPT Image-1's 63.07% and significantly higher than FLUX.1-dev's 47.49%. This highlights its exceptional ability to understand and render complex spatial and interactive relationships between multiple characters.
>   - **Strong Attribute Binding:** Its scores on "Character Attributes" (e.g., 49.11% for Animal, 39.30% for Person) are on par with or exceed those of previous SOTA models, indicating robust attribute binding even under high detail loads.
>   - **SOTA-Level Performance:** Qwen-Image's strong performance across all four dimensions validates that the *combination* of a powerful MLLM encoder (Qwen2.5-VL) and a robust diffusion backbone is a highly effective path toward mastering long-prompt generation. Nevertheless, *its scores remain far from perfect*, indicating considerable room for improvement.
>
> These new experiments **also underscore the unique contribution of our benchmark**. Simpler benchmarks focusing on short prompts might not reveal such a clear performance gap between models with different encoders. However, DetailMaster, with its average prompt length of over 284 tokens and its fine-grained evaluation across four critical dimensions, **provides the necessary complexity and granularity** to quantify the benefits of advanced encoders and identify persistent challenges. *It highlights that even with superior text understanding, the accuracy on the most difficult compositional dimensions* (Character Locations and Entity Relationships) *remains far from perfect.* In addition, handling intricate, long-form instructions remains an unsolved problem. We have newly added these new results and the discussion to the revised version of our paper (**Appendix O**).
>
> (To be continued.)

---

> ### Author Response · Authors · 2025-11-20
> **Responses to Reviewer 5wZt [Part 3/6]**
>
> (Continued from the previous page. W2 part1)
>
> ## **Evaluation on Unified Models**
>
> To provide a clearer picture of how recent unified models perform on our benchmark, we conduct an evaluation on several advanced unified models, including **Janus, JanusFlow, Janus-Pro, Lumina-Image-2.0, and BAGEL**. The results are presented in the table below:
>
> | Model                 | Object | Animal | Person | Character Locations | Background | Light | Style | Entity Relationships |
> | --------------------- | ------ | ------ | ------ | ------------------- | ---------- | ----- | ----- | -------------------- |
> | Janus-1.3B            | 35.86  | 32.15  | 22.31  | 21.4                | 87.59      | 95.67 | 83.53 | 25.48  |
> | JanusFlow-1.3B        | 31.93  | 37.54  | 16.5   | 17.3                | 81.24      | 94.8  | 92.99 | 16.77  |
> | Janus-Pro-1B          | 40.55  | 41.44  | 26.1   | 26.66               | 92.39      | 96.42 | 88.78 | 32.3                 |
> | Lumina-Image-2.0 (2B) | 43.74  | 43.79  | 29.2   | 34.28               | 86.86      | 93.47 | 93.81 | 39.68                |
> | BAGEL-7B-MoT          | 47.04  | 46.35  | 33.8   | 39.59               | 97.41      | 97.99 | 94.68 | 49.43                |
>
> Our analysis of these results reveals several insights into what drives performance on detail-rich, long-prompt generation:
>
> - **Janus-1.3B**: The original Janus model establishes a solid baseline. Its performance validates its core design of decoupling the understanding and generation encoders, which helps mitigate task conflict. Additionally, its unified architecture inherently enables it to comprehend longer prompts. However, **its capabilities are constrained by its 1.3B scale and initial training data**, leading to moderate scores in complex dimensions like Character Locations and Entity Relationships.
> - **JanusFlow-1.3B**: While JanusFlow achieves a very high Style score (suggesting high-quality image aesthetics), it underperforms the original Janus in most attribute-binding and spatial tasks. This result suggests that while the flow mechanism can improve overall image quality, **it may compromise the model's adherence to the fine-grained compositional details** prevalent in DetailMaster's prompts.
> - **Janus-Pro-1B**: Janus-Pro demonstrates a significant improvement across all metrics compared to its predecessors. As discussed in its paper, the "Pro" version **benefits from an optimized training strategy and substantially expanded training data**. This directly enhances its ability to interpret and render the complex compositional requirements of our benchmark, showcasing the critical role of data quality and training refinement in handling long prompts.
> - **Lumina-Image-2.0**: Lumina-Image 2.0 continues this upward trend. Its strong performance, especially the notable jump in Character Locations and Entity Relationships, can be attributed to its two key innovations mentioned in their work: **the Unified Next-DiT architecture and the Unified Captioner (UniCap)**. UniCap is specifically designed to produce high-quality, multi-granularity textual descriptions for T2I tasks. **Training on this highly descriptive data** aligns perfectly with the demands of our DetailMaster benchmark, enabling the model to better ground visual elements to detailed textual specifications.
> - **BAGEL-7B-MoT**: BAGEL achieves the highest performance across all dimensions. Its success stems from a combination of factors discussed in its paper: (1) **Model Scale:** At 7B parameters, it has a significantly larger capacity for knowledge and reasoning. (2) **MoT Architecture:** It employs a Mixture-of-Transformers (MoT) architecture with distinct experts for understanding and generation. This design minimizes task interference more effectively than a simple encoder decoupling, allowing each expert to specialize. (3) **Large-scale Interleaved Data:** Its training on trillions of tokens of interleaved multi-modal data equips it with superior compositional reasoning.
>
> In summary, these results demonstrate that performance on DetailMaster is strongly correlated with *architectural choices that mitigate task conflict* (e.g., decoupled encoders, MoT), *the richness of the training data* (e.g., UniCap), and *overall model scale*. This analysis further validates **the utility of our benchmark in discerning the key capabilities and limitations** of advanced unified models. We have newly added this discussion to the appendix of our paper (**Appendix P**).
>
> Furthermore, we have newly incorporated an analysis correlating the performance of these models with prompt length into **our revised paper (Appendix R, Figure 7, Figure 8)**. Although overall accuracy still declines with longer prompts, **these models demonstrate superior and more robust attribute alignment for generated characters**. This enhanced stability, especially for object and person attributes, confirms that these architectures are a promising optimization direction.
>
> (To be continued.)

---

> ### Author Response · Authors · 2025-11-20
> **Responses to Reviewer 5wZt [Part 4/6]**
>
> (Continued from the previous page. W2 part2)
>
> # W3: "many recent models actually leverage denser prompts during their training phase."
>
> Thank you for pointing out this. We acknowledge that our original statement was not sufficiently precise. Indeed, an increasing number of recent T2I models have begun to leverage longer and more detailed prompts during training.
>
> Our initial phrasing was based on the observation that **many classic and widely adopted T2I models** (e.g., those primarily using a CLIP text encoder) were trained under the constraint of CLIP's 77-token input limit. Most early datasets were constructed with this limitation in mind, so the majority of earlier models were trained on relatively short prompts. However, we fully agree that **recent research has increasingly recognized the importance of supporting longer prompts**. Many recent models have begun utilizing longer prompts during training and have adopted more powerful text encoders, such as T5 or various MLLMs.
>
> **We have revised the relevant sentences in our paper accordingly (particularly in the Instruction section).** Specifically, the original statement has been updated to: "`Classic` T2I models are trained on short-length prompts." This modification more accurately reflects the historical context while acknowledging recent advances.
>
>
> ---
>
>
> # Q1: A More Detailed Analysis of Flattening Complex Grammatical Structures and Attribute Leakage
>
> Thank you for pointing out the need for greater clarity regarding the analyses supporting  "flattening of grammatical structures" and "attribute leakage". We acknowledge that Section 4 does not contain explicitly titled subsections dedicated to these points, and we have newly addressed this by adding a dedicated subsection in the revised version to explicitly connect our experimental findings to these broader conclusions.
>
> In our original paper, the evidence supporting these two claims has been embedded within the following experimental results:
>
> - **Regarding encoders flattening complex grammatical structures (Evidence from Section 4.2.3 & Appendix M):** Our analyses in Section 4.2.3 and Appendix M show that "Character Locations" and "Entity Relationships" are among the most challenging tasks. These tasks require a nuanced understanding of *complex spatial instructions, inter-object relationships, and sentential hierarchies*. The models' failures in these areas strongly suggest that **their encoders struggle to parse such structured, non-linear descriptions**, instead **treating them as "flattened" sets of features**. Furthermore, our error analysis in Appendix M reveals dominant failure modes, including: (1) conflicts between spatial instructions and the model's real-world positional priors, and (2) difficulties in rendering complex character interactions. These failure cases directly reflect **the model's fundamental difficulty in comprehending instructions involving complex grammatical relations** (e.g., prepositional phrases, subordinate clauses).
> - **Regarding "attribute leakage" in diffusion models under detail-intensive conditions (Evidence from Section 4.3 & Appendix N):** One of our key findings is *the consistent decline in the accuracy of all evaluated models as prompt token length increases*. This is not solely due to character omission. As further detailed in Appendix N, **even for successfully generated characters, the fidelity of their attributes decreases as the prompt lengthens.** This is direct evidence of "attribute leakage": when the model is overwhelmed by a detail-intensive prompt, it fails to strictly "bind" specific attributes (e.g., colors, clothing, actions) to the correct characters, resulting in attributes being omitted, distorted, or incorrectly applied to other objects.
>
> We appreciate your feedback prompting us to clarify this. In the **revised manuscript**, we **have added a paragraph at the end of Section 4** to more explicitly link these specific experimental findings to our conclusions (**Section 4.4**).
>
> (To be continued.)

---

> ### Author Response · Authors · 2025-11-20
> **Responses to Reviewer 5wZt [Part 5/6]**
>
> (Continued from the previous page. Q1)
>
> # Q2.1: "How is the number of prompts changing from 4,565 to 4,116? How were they validated?"
>
> ## **Clarification on the Two-Round Data Construction Process**
>
> We would like to address a possible misunderstanding about our prompt enhancement method. **Our data construction process was carried out in two rounds.** In the first round, we used image data and detailed prompts from DOCCI and Localized Narratives as sources, and deployed Qwen2.5-14B-Instruct and Qwen2.5-VL-7B-Instruct for initial data synthesis. **The first round produced 4,565 image-text samples.** In the second round, we used these 4,565 samples as the original data and **reran the synthesis pipeline**, this time employing Qwen2.5-VL-72B-Instruct as both the LLM and MLLM. **This reprocessing step resulted in an improved dataset containing 4,116 refined prompts.** The reduction in the number of prompts from 4,565 to 4,116 is due to our validation and filtering procedures.
>
>
>
> ## **On the Validation Process for High-Quality Prompts**
>
> Regarding the validation of the prompts, as described in Sections 3.1.2 and 3.1.3, **we implemented multiple measures to ensure data quality**. These include: (1) deploying both the MLLM and an open-set object detection model to identify main characters and exclude those with ambiguous localization; (2) in the character attribute extraction step, using the MLLM to verify whether each extracted attribute genuinely belongs to the main character (answering "yes" or "no"), and discarding mismatched attributes; and (3) during the prompt refinement and filtering phase, validating whether each annotation aligns with the final prompt, and discarding those that do not match. These validation steps collectively ensured the high quality of the prompts and annotation data.
>
>
>
> ## **Human Evaluation as Validation of High Data Quality**
>
> Furthermore, in Section 5.2 and Appendix G of our original paper, **we present evaluations conducted by human experts** to assess the quality of our DetailMaster dataset. The results confirm that our dataset maintains high quality according to human judgment, which show that *100% of the annotations in the samples meet task relevance requirements, 93.6% maintain visual fidelity with source images, and 97.5% show complete consistency with final polished prompts*.
>
> **We have newly revised the relevant content of our paper** to make the data synthesis workflow clearer for readers (**L255-262**).
>
>
> ---
>
>
> # Q2.2: "...include all the system prompts that were used throughout this work."
>
> We appreciate the reviewer's attention to the methodological details of our prompt design. However, we would like to clarify that **all prompts used in this work** **have already been** **comprehensively documented in Appendix E of our original submission**. In addition, for full transparency and reproducibility, the prompts corresponding to each experimental step are also explicitly provided within our code repository.
>
> Regarding the specific system prompt for the LLMs (if the reviewer is referring to the identity-setting prompt), we consistently used the simple and standard formulation: **"You are a helpful assistant."** throughout all the steps.
>
> We hope this clarification adequately addresses the reviewer's concern.
>
>
> ---
>
>
> # Q4.1: "...what proportions of the samples from each dataset were used to construct the benchmark."
>
> We appreciate the reviewer's question regarding the sample proportions from each dataset. To clarify: **we utilized all available samples from both DOCCI and Localized Narratives (Flickr30k and COCO subsets)**, resulting in a total of **163K samples** initially.
>
> These samples then underwent a rigorous data synthesis and filtering pipeline. Specifically:
>
> 1. **Main character count filtering**: We retained only samples containing **4 to 8 main characters** to ensure that prompts have sufficient features while avoiding overly cluttered scenes.
> 2. **Minimum attribute filtering**: At each round of the data construction process, samples with **fewer than 4 valid character-level attributes** were removed.
>
> After applying these strict criteria, the initial 163K samples were reduced to 4K high-quality samples that form the final dataset.
>
> (To be continued.)

---

> ### Author Response · Authors · 2025-11-20
> **Responses to Reviewer 5wZt [Part 6/6]**
>
> (Continued from the previous page. Q4.1)
>
> # Q4.2: "...how the original captions aren't long enough and miss the important, desirable details?"
>
> Thank you for this insightful suggestion. We agree that providing quantitative analysis regarding the limitations of original captions would strengthen our discussion.
>
> Following your recommendation, we conduct a systematic analysis to *measure the rate at which original captions lack important fine-grained attributes*. Specifically, **we calculate which attributes from our final attribute list were missing in the original captions** and compute the corresponding proportions (termed as the **Attribute Lost Rate**).
>
> |                      | Attribute Lost Rate |
> | -------------------- | ------------------- |
> | Character Attributes | 63.07%              |
> | Character Locations  | 93.26%              |
> | Scene Attributes     | 60.20%              |
> | Entity Relationships | 68.36%              |
>
> As shown in the table, we find that:
>
> - The missing rate for **"Character Locations"** is as high as **93.26%**, indicating that positional information is rarely included. This quantitatively proves that original captions are **fundamentally insufficient** for training or evaluating precise control, validating the necessity of our prompt re-captioning pipeline. Our pipeline effectively addresses this by explicitly localizing characters and expanding the prompts with spatial context.
> - The missing rate for **"Entity Relationships"** is **68.36%**, suggesting that interactions between characters are frequently overlooked. Our method mitigates this by leveraging character identification and MLLM-based reasoning to extract and incorporate relational semantics.
> - The missing rate for **"Character Attributes"** is **63.07%**. Case studies reveal that these captions often include only a few salient attributes, lacking comprehensive and nuanced descriptions.
> - The missing rate for **"Scene Attributes"** (e.g., background, lighting, style features) is **60.2%**, showing that holistic scene characteristics are not consistently captured.
>
> **We have newly added this discussion, along with the corresponding quantitative results, in Appendix I of our revised paper as per your suggestion.** Thank you again for this valuable feedback.
>
>
>
> ---
>
>
>
> # Closing Remarks
>
> We want to express our deepest gratitude for the constructive suggestions and queries you provide. We dedicate a lot of time and effort to conducting additional experiments based on your recommendations, and we hope that you can consider an increase in the rating if these response with new results address your comments. If you have any additional concerns or queries, we respectfully invite you to share them with us. Thank you!

---

> > ### Comment · Reviewer_5wZt · 2025-11-21
> >
> > > we perform a new ablation on SD-XL that disentangles the effects of (i) increasing the token limit and (ii) explicitly training on long prompts.
> >
> > Thanks for this new result. Could the authors also discuss the performance of the fine-tuned model on short prompts -- if that gets negatively impacted? If so, does a scheme like RECAP [1] help?
> >
> > > MLLM-encoder-based diffusion models: SANA (Gemma2 encoder)
> >
> > Gemma2 doesn't accept multimodal inputs, right? Did the authors refer to PaliGemma2 [2]?
> >
> > > Lumina-Image-2.0 (2B)
> >
> > If the authors are referring to https://huggingface.co/Alpha-VLLM/Lumina-Image-2.0, I am not sure if it's a unified model, though. Could the authors clarify that?
> >
> > > These new experiments also underscore the unique contribution of our benchmark. Simpler benchmarks focusing on short prompts might not reveal such a clear performance gap between models with different encoders. However, DetailMaster, with its average prompt length of over 284 tokens and its fine-grained evaluation across four critical dimensions, provides the necessary complexity and granularity to quantify the benefits of advanced encoders and identify persistent challenges. It highlights that even with superior text understanding, the accuracy on the most difficult compositional dimensions (Character Locations and Entity Relationships) remains far from perfect. In addition, handling intricate, long-form instructions remains an unsolved problem. We have newly added these new results and the discussion to the revised version of our paper (Appendix O).
> >
> > Thanks for the additional clarification. I would advise the authors to include some results from these models (that use better text encoders) in the main text as well.
> >
> > > We have revised the relevant sentences in our paper accordingly (particularly in the Instruction section). Specifically, the original statement has been updated to: "Classic T2I models are trained on short-length prompts." This modification more accurately reflects the historical context while acknowledging recent advances.
> >
> > I would suggest making it even more specific by specifying the references to works that do and don't conduct training with long prompts.
> >
> > > On the Validation Process for High-Quality Prompts
> >
> > Could the authors discuss this a bit more in the main text / appendix? It would even be better if they could provide some examples after the filtering was successfully applied.
> >
> > Considering the discussions and analyses provided in the rebuttal, I have also raised my rating. Good luck.
> >
> > ## References
> >
> > [1] A Picture is Worth a Thousand Words: Principled Recaptioning Improves Image Generation; Segalis et al.; 2023.
> >
> > [2] PaliGemma 2: A Family of Versatile VLMs for Transfer; Steiner et al.; 2024.

---

> > > ### Author Response · Authors · 2025-11-23
> > > **New Responses to Reviewer 5wZt [Part 1/3]**
> > >
> > > Dear Reviewer 5wZt, we would like to express our sincere gratitude for your detailed review and thoughtful re-evaluation of our work. Your constructive engagement has greatly contributed to the improvement of our paper.
> > >
> > > Thank you also for providing six further points of clarification and suggestions for improvement. We have carefully considered each of them and have provided point-by-point responses below. In line with your valuable feedback, we have also made corresponding revisions throughout our paper.
> > >
> > > We hope that our responses and the accompanying modifications to the paper successfully address your remaining concerns and align with your excellent suggestions.
> > >
> > >
> > >
> > > ---
> > >
> > >
> > >
> > > # New Q1: Performance Robustness on T2I-CompBench (Short Prompts)
> > >
> > > Thank you for this insightful suggestion! To discuss the performance of the fine-tuned models in our ablation study on short prompts, we conduct a new evaluation using **T2I-CompBench**, a benchmark specifically designed for compositional T2I generation. The average prompt length in T2I-CompBench is significantly shorter, at only **12.65 tokens**, making it an ideal evaluation setting for short prompts.
> > >
> > > Our new evaluation rigorously assess prompt adherence across multiple dimensions, including: (1) **Attribute Binding:** color, shape, and texture; (2) **Character Location:** spatial relationships; (3) **Character Interaction:** non-spatial relationships; (4) **Complex Composition:** combination of multiple dimensions.
> > >
> > > The evaluation results of the SD-XL baseline and our four ablation models on T2I-CompBench are as follows:
> > >
> > > |                                    | Color      | Shape      | Texture    | Spatial    | Non-saptial | Complex    |
> > > | ---------------------------------- | ---------- | ---------- | ---------- | ---------- | ----------- | ---------- |
> > > | SD-XL                              | 0.5710     | 0.4741     | 0.5176     | 0.2035     | 0.3105      | 0.3239     |
> > > | 77 Limit w/ Short-Prompt-Training  | 0.5701     | 0.4712     | 0.5184     | 0.1890     | 0.3079      | 0.3214     |
> > > | 512 Limit w/ Short-Prompt-Training | 0.5862     | 0.4716     | 0.5168     | 0.1929     | 0.3074      | 0.3249     |
> > > | 77 Limit w/ Long-Prompt-Training   | 0.6057     | 0.4800     | 0.5417     | 0.2052     | 0.3105      | 0.3280     |
> > > | 512 Limit w/ Long-Prompt-Training  | **0.6108** | **0.4817** | **0.5608** | **0.2109** | **0.3112**  | **0.3395** |
> > >
> > > The results confirm that the performance of the fine-tuned models **does not suffer negative degradation on short prompts**.
> > >
> > > - **Short-Prompt Trained Models:** Models trained exclusively on short prompts (*77 Limit w/ Short-Prompt-Training* and *512 Limit w/ Short-Prompt-Training*) show marginal fluctuations compared to the SD-XL baseline, confirming that our short-prompt training process **does not harm general compositional capability**.
> > > - **Long-Prompt Trained Models:** The models trained using our detail-rich long prompts (*77 Limit w/ Long-Prompt-Training* and *512 Limit w/ Long-Prompt-Training*) exhibit **all-around improvements**. We observe **significant boosts in the Attribute Binding metrics (especially Color and Texture)**, suggesting enhanced object-attribute association capability. The *512 Limit w/ Long-Prompt-Training* model also **achieves the highest score in the Complex metric**, indicating superior compositional fidelity even when dealing with concise prompts.
> > >
> > > Crucially, these results **eliminate concerns regarding performance regression on short prompts**. They demonstrate that **training on compositionally complex data enhances the model’s overall compositional capability**, allowing it to adhere better to fine-grained constraints, **even in the context of concise prompts**.
> > >
> > > **We have newly incorporated this discussion and the new results table into Appendix S of our revised paper.** Thank you for helping us strengthen the analysis of our ablation study!
> > >
> > > (To be continued.)

---

> > > ### Author Response · Authors · 2025-11-23
> > > **New Responses to Reviewer 5wZt [Part 2/3]**
> > >
> > > (Continued from the previous page. New Q1)
> > >
> > > # New Q2: Correction of Text Encoder Categorization in the Supplementary Experiment
> > >
> > > Thank you for pointing out this issue! We appreciate the reviewer’s careful reading and agree that our previous statement was imprecise. SANA uses Gemma2 as its text encoder. As you noted, **Gemma2 is a pure text LLM** and does not support multimodal inputs.
> > >
> > > To address this ambiguity, we have revised the description in our supplementary experiment. Specifically, we now refer to this experiment section as "A Superior **LLM/MLLM** Encoder Yields Enhancements in Generative Output" to better distinguish between models using strong LLM encoders and those employing MLLM encoders. Within this clarified framework, **SANA leverages a stronger LLM encoder (Gemma2)**, whereas **Qwen-Image uses a stronger MLLM encoder (Qwen2.5-VL)**.
> > >
> > > We sincerely appreciate your meticulous feedback!
> > >
> > > ---
> > >
> > > # New Q3: Clarification on the Unified Nature of Lumina-Image 2.0
> > >
> > > Thank you for the comment. This model indeed corresponds to **the publicly released model "Lumina-Image-2.0" hosted at Alpha-VLLM’s repository on Hugging Face**. As described in its paper (i.e., *Lumina-image 2.0: A unified and efficient image generative framework*), Lumina-Image 2.0 is explicitly built upon a `unified architecture` known as **Unified Next-DiT**, which treats both text and image as a **joint token sequence** rather than processing them in separate modality-specific streams. In this design, text tokens and image latent tokens are concatenated into a single sequence and processed through a single-stream transformer using joint self-attention.
> > >
> > > Such a design conforms to the typical formulation of a **unified generative model**, where diverse modalities are encoded uniformly as tokens and jointly manipulated by the same backbone. This architecture supports seamless scaling and straightforward task extension because visual guidance prompts can be injected without modifying the core components. As emphasized in the Lumina-Image 2.0 paper, this unified structure not only improves image-text alignment but also serves as a foundation for for expanding the model to richer generative tasks without specialized architectural branches.
> > >
> > > Therefore, our use of the term *unified model* directly follows the model’s technical framework: Lumina-Image 2.0 treats **text and image tokens as a single coherent sequence for both representation learning and generation**, which is a hallmark of unified generative architectures.
> > >
> > > ---
> > >
> > > # New Q4: "... include some results from these models in the main text as well."
> > >
> > > Thank you for this valuable suggestion. **We have newly revised the paper according to your advice.** In the updated version, **we have moved the conclusions** of the experiments performed with stronger LLM/MLLM encoders and unified models **from the appendix to Section 4.2.2 in the main text**. This modification *allows readers to immediately access these results without referring to supplementary materials*.
> > >
> > > Specifically, **Section 4.2.2 now provides a clearer discussion** showing that *superior text encoders can improve the model’s ability to follow prompts, while unified architectures naturally enhance the model’s capacity for understanding long-form instructions*. In addition, the revised section reports the *supplementary findings* that both richer training data and larger model scales further contribute to performance gains under long-prompt settings.
> > >
> > > Furthermore, in addition to the expanded content in Section 4.2.2, **we have also newly included cross-references in lines 320–321 of Section 4.1 "Experimental Setup" and lines 512–514 of Section 6 "Further Analysis and Insights,"** directing readers to Appendix O and P for comprehensive experimental details.
> > >
> > > We sincerely appreciate your valuable comment, which has helped us improve the accessibility of our experimental conclusions.
> > >
> > > ---
> > >
> > > # New Q5: "... specifying the references to works that do and don't conduct training with long prompts."
> > >
> > > Thank you for the valuable comment. Following your suggestion, we have newly revised both the **Introduction** and **Related Works** sections in the new version of our paper.
> > >
> > > Specifically, in the first paragraph of the **Introduction**, we now explicitly list and reference representative models that are traditionally trained on short prompts, contrasting them with models that are specifically optimized for or trained on long prompts.
> > >
> > > Furthermore, **we have newly revised the paragraph titled "T2I models for long prompt"** within the **Related Works** section. The content has been restructured to more clearly associate each model with its respective training paradigm. This revision ensures that the distinction between short-prompt and long-prompt training is explicitly tied to the specific models being discussed, making the landscape of current research clearer for the reader. Thank you again for your valuable suggestion!
> > >
> > > (To be continued.)

---

> > > ### Author Response · Authors · 2025-11-23
> > > **New Responses to Reviewer 5wZt [Part 3/3]**
> > >
> > > (Continued from the previous page. New Q5)
> > >
> > > # New Q6: Details on the Validation Process for High-Quality Prompts
> > >
> > > Thank you for this valuable suggestion. As recommended, **we have newly added a section to the appendix of our revised paper** to provide *a more detailed discussion of our multi-stage validation process for ensuring high-quality prompts and annotations*, complete with *illustrative examples*.
> > >
> > > **The first validation step** in our pipeline focuses on *achieving accurate and unambiguous character localization*. To this end, **we employ a dual-model verification approach**, *generating two distinct bounding box proposals for each identified character using both an MLLM and the open-set detector YOLOE-11L*. We then reconcile these proposals based on their spatial agreement:
> > >
> > > - *If the two proposals have an Intersection over Union (IoU) of at least 0.7*, indicating strong consensus, we select the larger of the two boxes to ensure comprehensive coverage of the character. **For instance**, when tasked with localizing a "child wearing a gray hoodie," the proposals from YOLOE-11L ([121, 30, 357, 332]) and the MLLM ([135, 30, 364, 332]) yielded an IoU greater than 0.7. Therefore, we retained the larger YOLOE box.
> > > - *In cases where the IoU falls below 0.7*, we employ BLIP to generate image-text matching score for the content in each box against the character's name. If the scores of the two boxes are both lower than a threshold of 0.4, we consider that both options are incorrect. Otherwise, the higher-scoring box is then retained. **For example**, in localizing a "person with a bag walking away," the YOLOE result ([278, 59, 365, 291]) and the MLLM result ([198, 57, 304, 302]) had a low IoU. However, the MLLM's proposal achieved a higher BLIP score and the score was more than 0.4. Hence, it was selected as the definitive location.
> > >
> > > **Our second validation step**  ensures the fidelity of the extracted character attributes. **We iterate through every attribute associated with a character and use the MLLM to verify it** with a strict verification prompt: "*Please analyze the main character in this image. Is '{temp_characteristic}' one of its features? Only respond with 'yes' if it is a perfect match. Please only respond with 'yes' or 'no'.*" In this context, *temp_characteristic* represents the character attribute at the current iteration. If the MLLM responds with "no," we classify the attribute as a mismatch and discard it. This step is crucial for filtering out attributes that are ambiguous. **For example**, for a character described as a "child wearing a white shirt and blue shorts," the initially extracted attribute list was `['person', 'child', 'wearing blue jersey with number 12', 'short blonde hair', 'blue shorts', 'blue socks', 'playing soccer', 'white shirt']`. During verification, the MLLM determined that the number '12' on the jersey was **partially obscured, failing the "perfect match" criterion**. As a result, `the attribute "wearing blue jersey with number 12" was removed`, yielding a more accurate, filtered list: ['person', 'child', 'short blonde hair', 'blue shorts', 'blue socks', 'playing soccer', 'white shirt'].
> > >
> > > **The third validation step** takes place during the prompt refinement phase, **guaranteeing that our final annotation data is perfectly synchronized with the content of the final prompts.** We validate this alignment by instructing the MLLM to confirm whether each annotated attribute is explicitly described in the final prompt, using the guiding instruction: "*Your task is to determine whether the given image description includes the description of this particular CATEGORY feature of the main character.*" Here, *CATEGORY* denotes the current attribute category. Based on the MLLM's judgment, *any attributes not found in the final prompt are removed from the annotation set.* **For example**, for a character "beige labradoodle puppy sitting," the original attribute list was `['animal', 'labradoodle', 'puppy', 'sitting', 'fluffy', 'curly fur', 'soft expression']`. After the MLLM's review of the final prompt, it was determined that **the generic term "animal" and the subjective descriptor "soft expression" were not explicitly mentioned**. The final annotation list was therefore refined to `['labradoodle', 'puppy', 'sitting', 'fluffy', 'curly fur']`.
> > >
> > > **We have newly incorporated these detailed discussions and examples into Appendix T of our revised paper.** We hope this clarification adequately addresses your comment.
> > >
> > >
> > > ---
> > >
> > >
> > > # Closing Remarks
> > >
> > > Thank you for your valuable time and insightful comments! We have carefully considered all of your suggestions and hope that our responses and the corresponding revisions have fully addressed your concerns. We truly appreciate your support.

---

> > > > ### Comment · Reviewer_5wZt · 2025-11-24
> > > >
> > > > Yes, thanks for the detailed comments. I am satisfied with the rebuttal, and I wish you good luck.

---

> > > > > ### Author Response · Authors · 2025-11-25
> > > > >
> > > > > Dear Reviewer 5wZt, we are greatly encouraged to learn that our explanations and additional results have resolved your concerns. Your constructive comments have significantly helped us improve the quality and clarity of our paper.  Thank you very much for supporting our work!

---

### Author Response · Authors · 2025-11-26
**Kind Summarization of Paper Updates and Additional Experiments**

Dear Reviewers,

We sincerely thank you for your time and efforts in reviewing our paper, as well as the helpful feedback for this work! As the discussion period draws to a close, we would like to provide a **summary of the major revisions and additional experiments** we have conducted to **address all concerns raised by the reviewers**.

---

## **(1) New Model Evaluations**

**To demonstrate the generalizability of DetailMaster and analyze recent advancements**, we conduct evaluations on a wide array of advanced models:

- **Evaluation of Models with A Superior LLM/MLLM Encoder (5wZt W2, nKZ7 W1, YsLh W1.5):** We evaluate **SANA** (Gemma2 encoder) and **Qwen-Image** (Qwen2.5-VL encoder), and find that superior text encoders significantly boost prompt adherence, with Qwen-Image achieving performance competitive with proprietary models.

- **Evaluation of Unified Models (5wZt W2, YsLh W1.4):** We assess **Janus, JanusFlow, Janus-Pro, Lumina-Image-2.0, and BAGEL**. The results show that unified architectures improve model adherence to long, complex prompts. Data richness and model scale are further key to advancement.

---

## **(2) Rigorous Controlled Ablation Study**

We perform **a controlled ablation using a modified SD-XL architecture** (incorporating T5) under four settings to isolate the effects of **prompt token limit** and **training data length**:

- **Controlled Ablation Study (5wZt W1 & Q3, YsLh W1.2):** We conduct a rigorous ablation study by equipping SD-XL with a T5 encoder and training under different constraints (77 vs. 512 tokens; short vs. long training prompts). The results confirm that **both** an expanded prompt limit and long-prompt training **are essential for performance, with a clear synergistic effect**.
- **Validation on Short Prompts (5wZt New Q1):** We evaluate our ablation models on T2I-CompBench (short prompts). The results show that our long-prompt training strategy **improves compositional capabilities even on short prompts**, ensuring no performance regression.

---

## **(3) Evaluator Robustness & Accessibility**

To address concerns regarding the use of MLLMs as evaluators:

- **Cross-Model Validation (nKZ7 W4, PbGa W2.2):** We re-evaluate the Qwen-based model using **InternVL3-9B** to check for self-enhancement bias, and find that relative rankings remain consistent.
- **Lightweight Evaluator Support (YsLh Q, PbGa W2.2):** We validate the evaluation pipeline using the smaller **Qwen2.5-VL-3B-Instruct**. The strong correlation with the 7B model confirms that DetailMaster is accessible to researchers with limited computational resources.

---

## **(4) Enhanced Analysis and Clarifications**

Beyond the additional experiments, we resolve specific methodological concerns:

- **Attribute Leakage & Structure Flattening (5wZt Q1, YsLh W1.1):** We add a dedicated analysis explaining how encoders "flatten" complex grammatical structures and how diffusion models suffer from "attribute leakage" under high detail loads.
- **The Role of Source Images & Data Validation (nKZ7 W3, PbGa W2.1 & Q1):** We clarify that our validation pipeline ensures the high quality of the final prompts. These prompts are used as the ground truth for evaluation, while the source images function solely as seeds.
- **The Need for Our Prompt Re-captioning Pipeline (5wZt Q4):** We introduce the "Attribute Lost Rate" metric and conduct a new quantitative analysis to demonstrate the limitations of original captions, providing strong justification for our data synthesis pipeline.
- **Robust Character Disambiguation (nKZ7 W2):** We clarify that characters are identified using unique descriptive modifiers, which ensures accurate localization and attribute binding even in crowded scenes.

---

The discussion phase will end in a few days and we have received only one response. We believe the paper has been further improved with your constructive comments.

We have carefully addressed each of the points raised and have made corresponding revisions to the paper. We kindly invite you to review our responses and reconsider the merits of our work. Should you have any further questions or require clarification, we warmly welcome your input. Your feedback is highly valued and appreciated. Thanks again!



Best, authors

---

### Author Response · Authors · 2025-12-01
**Kind Summarization of Author-Reviewer Discussion**

Dear Area Chair, Senior Area Chair, and Program Chairs,

We sincerely appreciate the time and effort you dedicate to meta-reviewing our paper. We understand that the current situation significantly increases your workload. To assist you as much as we can, we provide a summary of our author-reviewer discussion.

During the author-reviewer discussion period, we have **thoroughly addressed all the reviewers' questions and misunderstandings**, and conducted **several new experiments** based on their suggestions ([paper updates](https://openreview.net/forum?id=AZ0VQouDmR&noteId=FgPE8ZQ5KE)). All of these new experiments and discussions **have been incorporated into our revised paper**.

- **Reviewer 5wZt** raised their score from **4 to 6** [Confidence 5] following our **two-round discussion**. Notably, this positive response and score update were received **well before 11.26 AoE (to our knowledge, the time of the OpenReview information leak)**. In our **initial response**, we conducted `a new ablation study` that *systematically isolated the impacts of token limit and training data length* (Part 1/6). We also `expanded our evaluation` to more advanced models to *analyze the connection between performance and optimization methods* (Part 2-3/6); and introduced the "Attribute Lost Rate" metric with `a new quantitative analysis` to *provide strong justification for our prompt re-captioning pipeline* (Part 6/6). **The reviewer responded positively within half a day, raising their score to 6** and providing follow-up questions. In our **second-round response**, we `evaluated our ablation models on T2I-CompBench (short prompts)` (New Part 1/3), with results showing that *our long-prompt training strategy improves compositional capabilities even on short prompts*. We also `incorporated all of their suggested revisions` into our paper (Other Parts).
- In response to **Reviewer PbGa (Score: 6)**, we **addressed all raised concerns**, though the discussion period concluded before we could receive a follow-up. Following the reviewer's suggestion, we `incorporated new related work` into our discussion (Part 1/2). Furthermore, we `provided a detailed clarification` on *the permissibility of "hallucinations" in our data generation process* (Part 1/2) and *the methods we employ to ensure prompt reasonableness* (Part 2/2). We also `supplied evidence` for *the robustness and reproducibility of our MLLM-based evaluation*.
- For **Reviewer nKZ7 (Score: 4)**, we **conducted extensive new experiments** to address their questions, though the deadline for reviewer replies passed before we received a follow-up. We evaluated `SANA` and `Qwen-Image`, finding that *superior text encoders significantly boost prompt adherence* (Part 1/4). We also `detailed our method` for achieving *robust disambiguation of interactive repeated characters* (Part 2/4), `clarified the role of source images versus final prompts` in our benchmark (Part 3/4), and `re-evaluated the Qwen-based model using InternVL3-9B` to demonstrate that our results are *not subject to self-enhancement bias* (Part 4/4).
- For **Reviewer YsLh (Score: 4)**, we **addressed all points raised** but the discussion period ended before a follow-up could be received. We `assessed a suite of advanced unified models`, showing that *unified architectures, rich data, and model scale are key to mastering long prompts* (Part 5/7). We `added a dedicated analysis` explaining *how encoders "flatten" grammatical structures and how diffusion models suffer from "attribute leakage"* (Part 1/7). We also `elaborated on the reasons` for *diverse model behaviors* (Part 2-3/7) and *how DetailMaster better diagnoses long-prompt performance than prior benchmarks* (Part 4/7). To `enhance accessibility`, we validated our entire pipeline using *a smaller evaluator* (Part 7/7).

**We are greatly encouraged by the remarks from the reviewers**, such as “existing benchmarks are indeed lacking in this aspect” and “conducts extensive experiments” by **Reviewer nKZ7**; “well-written and easy-to-follow” and “the experiments are extensive” by **Reviewer PbGa**; and “reveals critical T2I flaws and guides research”  by **Reviewer YsLh**. We are especially motivated by **Reviewer 5wZt**'s positive feedback, including their comments on the "Robustness and validity of the benchmark" and their concluding remark: **“Thanks for the detailed comments. I am satisfied with the rebuttal.”**

In short, the ratings up to 11.26 AoE are **(4→6) [Conf. 5], 6 [Conf. 4], 4 [Conf. 4], and 4 [Conf. 3]**, with **three reviewers not having responded**.

If you have any further questions or suggestions regarding DetailMaster, we would be happy to continue the discussion before the discussion period ends. If possible, having a bit of advance notice would also help us prepare a thorough response. Thank you again for your efforts in the review process and for the opportunity to engage with these excellent reviewers.

Best, authors

---

### Note · Authors · 2026-01-29

I have read and agree with the venue's withdrawal policy on behalf of myself and my co-authors.

---

### Meta-Review · Area_Chair_CxzM · 2025-12-24

**Summary:**

1) Lack of controlled experiments to drive the insights and analyses in Section 4
2) Lack of results with several models that are known for handling long and complex prompts
3) During evaluation, DetailMaster needs to detect the bounding box for each character based on the Character List. The data generation protocol seems limited to certain types of edits.
4) Newer models that can handle longer prompts are not adequately considered
5) Discussion of related work
6) Incomplete Failure Mechanism Validation

**Reviewer Concerns:**

The reviewers partially addressed the concerns, but overall, it is hard to be convinced that there is enough contribution. The insights from the paper are not especially surprising, and the synthetic data generation pipeline is also borderline.

**Reviewer Scores:**

The scores are predicted to be 4, 4, 6, 6 after the rebuttal.

---

### Decision · Program_Chairs · 2026-01-26

Reject